# Adversarial Reinforcement Learning for Robust Diffusion Large Language Model Unlearning

**Zhiwei Zhang** [1]   **Yudi Lin** [2]   **Linlin Wu** [3]   **Fali Wang** [1]   **Yi Xin** [1]   **Xiaomin Li** [4]   **Minhua Lin** [1]   **Xianfeng Tang** [5]   **Qi He** [5]   **Suhang Wang** [1]

## Abstract

Diffusion language models (DLMs) have recently emerged as an alternative to autoregressive approaches, enabling parallel sequence generation and flexible token generation orders. Machine unlearning plays a critical role in mitigating legal and ethical risks by removing the influence of specific training examples from trained models. While unlearning has been extensively studied for autoregressive language models, its applicability to DLMs remains unexplored. The architectural differences of DLMs raise new challenges for effective and robust unlearning that are not addressed by existing methods. In this paper, we present the first comprehensive study of unlearning for DLMs. Through systematic empirical analysis, we show that unlearning performance in DLMs is highly sensitive to generation hyperparameters, highlighting the need for evaluation across diverse generation settings. We further observe that DLMs tend to reproduce unlearned information when target inputs are embedded within informative contexts, due to their ability to incorporate both prefix and suffix conditioning, which increases vulnerability to elicitation attacks and weakens the robustness of existing unlearning methods. To design a robust unlearning method, we propose an adversarial reinforcement learning framework, where a context generator adversarially produces informative contexts to elicit unlearned knowledge, while the DLM is optimized to suppress undesired recall. We further introduce novel components to address credit assignment and stability issues in this adversarial learning setup. Extensive experiments show that our method significantly improves unlearning

effectiveness while preserving model utility.

## 1. Introduction

Masked diffusion models are a promising and scalable architecture for diffusion language models (Nie et al., 2025; Ye et al., 2025a), enabling massive inference acceleration through parallel generation (Labs et al., 2025; DeepMind, 2024; Song et al., 2025) and improved consistency via bidirectional attention (Ye et al., 2024a; Zhang et al., 2023). DLMs have been applied to coding (Labs et al., 2025; DeepMind, 2024; Gong et al., 2025; Xie et al., 2025) and reasoning tasks (Ye et al., 2024a; 2025a), while block-attention variants (Arriola et al., 2025; Cheng et al., 2025) have been scaled to excel at complex reasoning.

Large language models are often trained on massive corpora that may contain harmful, sensitive, or copyrighted content (Yao et al., 2024a; Wen et al., 2023; Li et al., 2024; Hendrycks et al., 2023). Language model unlearning has therefore emerged as an important mechanism for removing the influence of specific training examples or capabilities from pretrained models without retraining from scratch (Zhang et al., 2025c; Liu et al., 2025b; Wang et al., 2025a; Fan et al., 2025; Shen et al., 2025; Zhang et al., 2025b; Geng et al., 2025; Maini et al., 2024; Shi et al., 2024b; Yao et al., 2024a; Dorna et al., 2025; Li et al., 2025d). While unlearning has been extensively studied for autoregressive language models, its applicability to diffusion-based language models (DLMs) remains unexplored. Extending unlearning from autoregressive LLMs to diffusion language models is not a trivial adaptation, as DLMs rely on fundamentally different generation mechanisms, including bidirectional conditioning, recurrent denoising, and parallel full-sequence generation. These differences introduce new challenges for effective and robust unlearning.

In this paper, we present the first comprehensive study of unlearning for diffusion language models. We begin with a systematic empirical analysis demonstrating that unlearning performance in DLMs is highly sensitive to generation hyperparameters, particularly the output length and the number of denoising steps. These results highlight the necessity

---

[1]The Pennsylvania State University [2]Nanyang Technological University [3]University of Utah [4]Harvard University [5]Microsoft. Correspondence to: Zhiwei Zhang <zbz5349@psu.edu>.

*Proceedings of the $43^{rd}$ International Conference on Machine Learning*, Seoul, South Korea. PMLR 306, 2026. Copyright 2026 by the author(s).

of evaluating unlearning under diverse generation configurations, rather than relying on a single inference setting.

Beyond hyperparameter sensitivity, we identify a critical robustness issue that is specific to diffusion language models. Unlike autoregressive models, DLMs support more flexible conditioning patterns, including prefix–suffix conditioning as well as interleaved mask–text infilling. Recent work has shown that this flexibility makes diffusion models vulnerable to jailbreak attacks via carefully crafted interleaved mask–text prompts (Wen et al., 2025). However, it remains unclear whether such vulnerabilities persist after the model is explicitly fine-tuned to resist adversarial contexts, and whether existing unlearning methods can robustly suppress knowledge recall under previously unseen contextual perturbations. Through extensive experiments, we show that even when DLMs are unlearned with adversarial contexts during training, they remain vulnerable to unseen adversarial contexts at inference time. In particular, unlearning performance under such adversarial contexts can degrade to a level comparable to that of the original, non-unlearned model evaluated on clean inputs. These findings suggest that effective unlearning in diffusion language models requires robustness beyond fixed adversarial perturbations observed during training, and motivate the need for stronger and more adaptive unlearning strategies.

Based on our preliminary experiments, we propose an adversarial reinforcement learning framework for robust unlearning in diffusion language models. Our method jointly trains a DLM and an adversarial context generator in a min–max game: the generator produces prefix–suffix contexts that aim to induce undesired recall of the target response, while the DLM is optimized to suppress such recall under these contexts. However, such an adversarial unlearning framework faces several nontrivial challenges in diffusion language models, including (i) ambiguous credit assignment between adversarial prefixes and suffixes, and (ii) instability caused by adversary–learner imbalance, where the adversary generates contexts that are either too difficult or too easy for the current unlearning model, leading to reward collapse during training.

To address these challenges, we introduce two key design components. First, we incorporate a causal influence regularization that explicitly quantifies the contribution of prefixes and suffixes to the model's predictions, enabling fine-grained credit assignment and preventing the generator from converging to solutions dominated by a single component. Second, we propose a regret-based difficulty calibration mechanism that compares the current unlearning policy against a stronger self-bootstrapped reference model, encouraging the generator to produce adversarial contexts that are neither trivially ineffective nor excessively informative. Together, these mechanisms stabilize adversarial training

and significantly improve the robustness of unlearning under previously unseen adversarial contexts.

Our **main contributions** are as follows: **(1)** We provide the first comprehensive study of unlearning for DLMs, systematically characterizing their unlearning behavior. **(2)** We show that unlearning performance in DLMs is highly sensitive to generation hyperparameters and remains fragile under adversarial contexts, revealing fundamental robustness limitations of existing unlearning approaches. **(3)** We propose a novel adversarial reinforcement learning framework for DLM unlearning, which integrates causal influence regularization and regret-based difficulty calibration to achieve robust unlearning performance.

## 2. Related works

### 2.1. Diffusion Large Language Models

Diffusion language models (DLMs) have emerged as a promising alternative to autoregressive language models, enabling parallel generation through iterative denoising (Austin et al., 2021; Nie et al., 2025; Sahoo et al., 2024; Lou et al., 2023; Gat et al., 2024; Shi et al., 2024a). Early work on discrete DLMs defines the diffusion process directly on token vocabularies with structured transition matrices and simplified training objectives (Austin et al., 2021; He et al., 2023; Zheng et al., 2023; Shi et al., 2024a; Sahoo et al., 2024; Ou et al., 2024; Lou et al., 2023; Gat et al., 2024). Recent efforts have scaled discrete DLMs to billions of parameters. LLaDA (Nie et al., 2025) demonstrates that discrete DLMs can be trained from scratch at 8B scale, achieving performance comparable to LLaMA3-8B, with LLaDA 1.5 (Zhu et al., 2025a) further improving alignment through preference optimization. Dream (Ye et al., 2025a) and DiffuLLaMA (Gong et al., 2024) show that discrete DLMs can be effectively adapted from pretrained AR models. Further developments include long-context extensions (Liu et al., 2025c; He et al., 2025), mixture-of-experts architectures (Zhu et al., 2025b), hybrid AR-diffusion approaches (Arriola et al., 2025; Cheng et al., 2025; Liu et al., 2025a), code generation (Gong et al., 2025), and chain-of-thought reasoning (Ye et al., 2024b). Industry efforts such as Mercury (Labs et al., 2025), Gemini Diffusion (DeepMind, 2024), and Seed Diffusion (Song et al., 2025) report inference speeds of thousands of tokens per second. DLMs have also been extended to multimodal settings, enabling unified understanding and generation across text and images (You et al., 2025; Li et al., 2025b; Yu et al., 2025; Yang et al., 2025; Wang et al., 2025b; Shi et al., 2025; Li et al., 2025c; Swerdlow et al., 2025; Xin et al., 2025b; Li et al., 2025a; Tian et al., 2025; Xin et al., 2025a). However, unlearning for DLMs remains unexplored.

## 2.2. Large Language Model Unlearning

Machine unlearning for LLMs (Yao et al., 2024b; Zhang et al., 2024; Liu et al., 2024; Jin et al., 2024; Łucki et al., 2024; Ji et al., 2024; Fan et al., 2024; Yuan et al., 2025; Shen et al., 2025; Jia et al., 2024) has attracted increasing attention due to concerns over privacy, copyright, and safety risks (Geng et al., 2025; Liu et al., 2025b). Existing methods can be broadly categorized into: (1) *gradient-based approaches*, such as Gradient Ascent (Jang et al., 2023) and Gradient Difference (Maini et al., 2024), which increase loss on the forget set; (2) *preference optimization methods*, such as NPO (Zhang et al., 2024), which reformulate unlearning as preference learning; and (3) *localization-based methods*, such as RMU (Li et al., 2024), which modify specific parameters storing target knowledge. Benchmarks including TOFU (Maini et al., 2024), WMDP (Li et al., 2024), and MUSE (Shi et al., 2024b) have been proposed to evaluate unlearning effectiveness and utility preservation. Unlearning for DLMs faces new challenges and remains unexplored.

## 2.3. DLM Reinforcement Learning

Reinforcement learning plays a central role in aligning LLMs with human preferences and improving reasoning ability. While RLHF (Ouyang et al., 2022) with PPO (Schulman et al., 2017) demonstrated this potential, PPO relies on a separate critic network and incurs substantial overhead. GRPO (Guo et al., 2025) removes this dependency and computes advantages via group-wise comparisons, and it has been widely adopted (Guo et al., 2025; Zhang et al., 2025a; Liu et al., 2025d; Zheng et al., 2025). Extending RL to DLMs presents unique challenges since log-likelihoods are intractable in iterative denoising. Recent solutions include diffu-GRPO (Zhao et al., 2025) with mean-field decomposition, coupled-GRPO (Gong et al., 2025) with complementary mask sampling, SEPO (Zekri & Boullé, 2025) based on score entropy, and UniGRPO (Yang et al., 2025) with structured noising for multimodal DLMs. For preference optimization, VRPO (Zhu et al., 2025a) reduces ELBO variance via optimal sampling allocation. Beyond policy optimization, DoT (Ye et al., 2024b) and DCoLT (Huang et al., 2025) explore parallel reasoning chains.

## 3. Preliminaries

### 3.1. Unlearning Method

Specifically, we adopt the diffusion-based language model `GSAI-ML/LLaDA-8B-Instruct` (Nie et al., 2025) as our baseline. We first fine-tune the model on the *forget* split of the TOFU dataset (Maini et al., 2024), which is designed for fictitious unlearning in large language models. This warm-up fine-tuning step ensures that the diffusion language model captures task-specific knowledge that may

not be fully covered during the pretraining of LLaDA. Following (Nie et al., 2025), the training objective is:

$$\mathcal{L}(\theta) = -\mathbb{E}_{t, p_0, r_0, r_t} \left[ \frac{1}{t} \sum_{i=1}^{L} \mathbf{1}\left[r_t^i = \mathrm{M}\right] \log p_\theta\left(r_0^i \mid p_0, r_t\right) \right],$$
(1)

where $t \sim \mathrm{Uniform}(0, 1)$ is a continuous random variable controlling the masking rate, $p_0$ denotes the input prompt, $r_0$ is the corresponding response, and $r_t$ represents a partially masked version of $r_0$, in which each token is independently masked with probability $t$. The indicator function $\mathbf{1}[r_t^i = \mathrm{M}]$ restricts the loss to masked positions, encouraging the model to reconstruct the original response tokens conditioned on the prompt and the corrupted response.

Given the fine-tuned baseline diffusion language model, we next perform unlearning. To improve robustness against extraction-based attacks, we incorporate a generator model that constructs adversarial contextual perturbations.

Specifically, we use a language model as the context generator and explicitly prompt it to take the original prompt $p_0$ and response $r_0$ as input and produce an adversarial prefix $\pi$ and a suffix $\sigma$. Detailed prompts are provided in Appendix A.3. For non-QA datasets, we convert each text sequence into a pseudo QA format by splitting it into a prompt segment and a target response segment. The resulting prefix–suffix contexts are designed to provide relevant background information that facilitates recall of $r_0$, while preventing the target model from trivially inferring $r_0$ from local context alone. The resulting input to the diffusion language model is formed as:

$$\tilde{r}_t = \pi \oplus p_0 \oplus r_t \oplus \sigma,$$
(2)

where $\oplus$ represents sequence concatenation. Following the idea of gradient ascent for unlearning in autoregressive language models (Jang et al., 2023; Yao et al., 2024b), the **unlearning objective** for the diffusion language model is defined as:

$$\mathcal{L}(\theta) = \mathbb{E}_{t, p_0, r_0, r_t} \left[ \frac{1}{t} \sum_{i=1}^{L} \mathbf{1}\left[r_t^i = \mathrm{M}\right] \log p_\theta\left(r_0^i \mid \tilde{r}_t^i\right) \right],$$
(3)

The objective is optimized such that the diffusion language model is encouraged to forget the target knowledge in $r_0$ even when the prompt is augmented with informative prefixes and suffixes. More training details are in Appendix A.5.

### 3.2. Evaluation Protocol

We then benchmark both the baseline and the unlearned models to evaluate the *effectiveness of unlearning* and the *preservation of model utility*.

As diffusion-based language models require explicit specification of both the output length and the number of

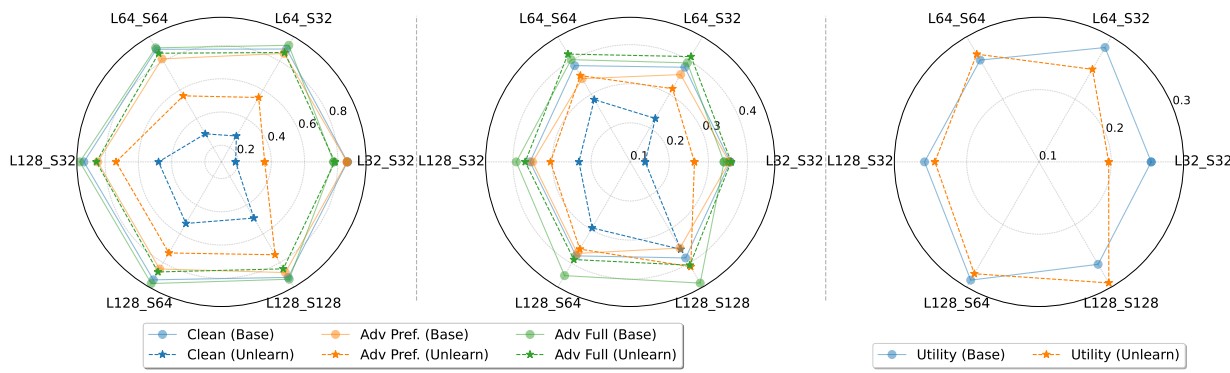

*Figure 1.* Unlearning performance of the base and unlearned models across six hyperparameter settings, where $L$ denotes the output length and $S$ denotes the number of denoising steps. **Left:** Unlearning effectiveness evaluated by LLM-as-Judge. **Middle:** Unlearning effectiveness measured by ROUGE. **Right:** Model utility. Lower is better for the unlearning performance, while higher is better for utility.

denoising steps at inference time, we report results under six configurations corresponding to different combinations of sequence length $L$ and diffusion steps $S$: $(L, S) = (32, 32), (64, 32), (128, 32), (64, 64), (128, 64), (128, 128)$. We set the minimum sequence length to 32, as responses in the unlearning dataset are typically short.

We evaluate performance along two dimensions: *unlearning effectiveness* and *model utility*. For unlearning effectiveness, we report ROUGE scores, which measure the degree of overlap between the model-generated responses and the ground-truth answers. In addition, we employ an LLM-as-Judge evaluation protocol, where a separate large language model is used to assess whether the generated outputs still contain information corresponding to the forgotten data. Further implementation details on the model and prompt construction are described in the Appendix A.3.

To thoroughly evaluate the robustness of unlearning, we employ a context generator to produce an adversarial prefix $\pi$ and suffix $\sigma$ following the setting in Sec. 3.1. Based on the generated contexts, we construct three input variants for evaluating the unlearned model: (1) **Clean:** the input consists of the original prompt $p_0$. (2) **Adversarial Prefix:** the input is constructed as $\pi \oplus p_0 \oplus r_t$. (3) **Adversarial Full:** the input is constructed as $\pi \oplus p_0 \oplus r_t \oplus \sigma$.

To assess model utility, we follow the evaluation protocol established in (Maini et al., 2024; Dorna et al., 2025), which measures both utility on retain dataset and general utility on out-of-distribution or general-purpose benchmarks. The specific evaluation protocol is detailed in Appendix A.4.

### 3.3. Results analysis

The results are reported in Fig. 1. We make the following observations:

*(1) Sensitivity to hyperparameter settings.* As shown in Fig. 1(a), unlearning performance exhibits substantial vari-

ance across hyperparameter settings under the LLM-as-Judge evaluation and ROUGE scores. In general, longer output lengths are associated with a higher risk of knowledge recall, indicating that the DLM becomes increasingly vulnerable as the generation length increases. In contrast, the effect of the number of denoising steps is less consistent. For example, when the output length is fixed at 128, the unlearning performance with 32 denoising steps is comparable to that with 128 steps, while outperforming the configuration with 64 steps under the LLM-based judge metric. This suggests that denoising steps do not induce a monotonic or easily predictable impact on unlearning effectiveness.

*(2) Impact of contextual variants.* The base model exhibits relatively consistent unlearning performance across the three input variants. In contrast, the unlearned model achieves significantly improved unlearning effectiveness compared to the base model under clean inputs, but its performance degrades progressively from clean inputs to prefix-only and adversarial full contexts. This trend indicates that the DLM remains sensitive to informative contextual cues, with more informative contexts leading to stronger knowledge recall.

*(3) Limited robustness under adversarial contexts.* Even the model utility is largely preserved after unlearning, the unlearning performance of the unlearned model under the adversarial full setting is comparable to that of the base model evaluated under clean inputs. This result highlights a key limitation: although the DLM is trained with adversarial contexts during unlearning, it still lacks robustness to previously unseen contexts at evaluation time.

Based on our comprehensive preliminary experiments, we draw the following conclusions:

> (1) Unlearning evaluation for diffusion language models should cover different settings of output length and denoising steps.

(2) Diffusion language models require more robust unlearning against potential extraction attacks via adversarial prefixes and suffixes.

## 4. Methodology

Our preliminary experiments show that although DLM is trained with adversarial prefixes and suffixes to improve robustness against extraction attacks, its unlearning effectiveness remains limited when evaluated under previously unseen contexts. This suggests that robustness induced by fixed adversarial perturbations during training does not necessarily generalize to broader contextual variations at evaluation time. A straightforward idea is to introduce a context generation mechanism during unlearning, which produces multiple contextual variants for each prompt. However, enforcing robustness over a wide range of adversarial contexts may negatively impact model utility, a trade-off that has been consistently reported in prior studies (Zhang et al., 2025c; Liu et al., 2025b; Wang et al., 2025a; Fan et al., 2025). Supervised fine-tuning optimizes the model by matching a specific target output, providing a clear supervision signal for learning desired behaviors. In unlearning, however, supervised objectives no longer define a positive target and instead impose a negation constraint: the model is trained to avoid producing the forgotten content, while any alternative output is treated as acceptable. This weak semantic guidance can induce uncontrolled distributional drift and lead to substantial utility degradation (Shi et al., 2024b; Yao et al., 2024a; Dorna et al., 2025; Li et al., 2025d).

Reinforcement learning follows a different optimization paradigm by operating over multiple candidate responses with relative reward signals. Given a prompt and a set of candidate responses with associated rewards, policy updates reweight the output probability mass toward higher-reward responses while suppressing lower-reward ones. This preference-based reweighting enables more controlled behavior shaping during unlearning and avoids arbitrary deviations from the desired output distribution (Guo et al., 2025; Wu & Choi, 2025), which is verified in Sec. 5.

Motivated by this property, we propose an adversarial reinforcement learning framework for DLM unlearning that aims to improve robustness to adversarial contexts while mitigating utility collapse. In this framework, a context generator is trained to produce contexts that increase the likelihood of undesired recall, while the DLM is optimized to suppress the corresponding target outputs under these contexts. Both the context generator and the DLM are trained via reinforcement learning with opposing reward signals, forming a min–max optimization process. Specifically, given a context generator $f_c$ and a DLM $\pi_\theta$, for each

prompt $p_i$ with corresponding response $r_i$, the context generator produces $K$ context variants consisting of prefix–suffix pairs $\{(\alpha_i^k, \beta_i^k)\}_{k=1}^K$. Each variant defines a new prompt

$$\tilde{p}_i^k = \alpha_i^k \oplus p_i \oplus [\text{MASK}] \oplus \beta_i^k.$$

For each $\tilde{p}_i^k$, we sample $G$ predictions from the DLM on the masked region, yielding generated outputs $\{y_i^{k,j}\}_{j=1}^G$. We introduce a judge model to assess whether a generated prediction $y_i^{k,j}$ matches the reference response $r_i$. For each prediction, a binary reward is assigned as

$$R(y_i^{k,j}) = \begin{cases} +1, & \text{if } y_i^{k,j} \text{ does not match } r_i, \\ -1, & \text{otherwise.} \end{cases} \quad (4)$$

However, this adversarial reinforcement learning framework introduces several nontrivial challenges: *(1) Credit assignment between prefix and suffix generation.* The adversarial context generator produces both a prefix and a suffix for each prompt, while receiving a scalar reward based solely on attack success or failure. Under this formulation, the reward signal does not provide explicit credit assignment between the prefix and suffix components. As a result, it is unclear whether both components contribute meaningfully to facilitating knowledge recall, or whether the observed effect is dominated by one of them. This ambiguity can induce an imbalanced learning dynamic, causing the generator to converge to solutions dominated by a single component, thereby limiting the effectiveness of attacks in cases where both components jointly contribute to successful extraction, which is verified in Sec. 5.3. *(2) Adversary–learner imbalance and reward signal collapse.* A second challenge arises from the imbalance between the adversarial context generator and the unlearning policy. When the generator is overly strong, or fails to strictly adhere to the intended constraint of providing only auxiliary background information (rather than implicit hints or direct inference cues), it may produce contexts that trivially facilitate the recovery of the ground-truth content. In the early stages of training, when the DLM has not yet developed robustness to adversarial contexts, most rollout trajectories may therefore result in predictions that exactly match the ground-truth responses. As a consequence, the reward signal becomes nearly uniform and negative, leading to vanishing reward variance and ineffective policy updates. A symmetric failure mode occurs when the generated contexts are overly weak or uninformative. In this case, the DLM consistently avoids producing the ground-truth responses, resulting in uniformly positive rewards across trajectories. Ideally, effective adversarial unlearning requires a balanced training regime in which the context generator produces contexts of intermediate difficulty. Such a regime preserves a mixture of positive and negative rewards across rollout trajectories, thereby maintaining sufficient reward variance to provide informative gradients for policy optimization. In contrast,

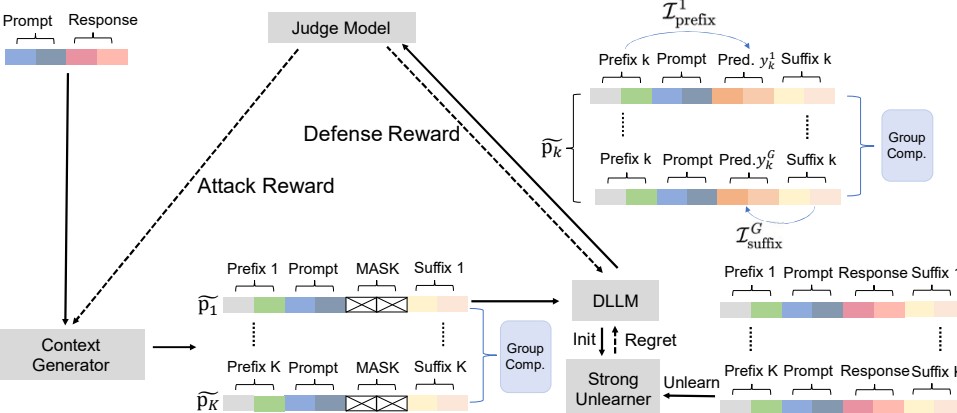

*Figure 2.* The framework of our method. The DLM $\pi_\theta$ and a context generator $f_c$ are trained adversarially via reinforcement learning. The generator produces adversarial prefix–suffix contexts to induce undesired recall, while the DLM learns to suppress the target response. A judge model assigns a binary reward by evaluating whether the DLM output matches the ground-truth response $r_i$ to be unlearned. Causal influence regularization enables prefix–suffix credit assignment, and a regret-based signal calibrates the difficulty of generated contexts.

contexts that are either overly challenging or overly trivial lead to degenerate reward signals and undermine stable and effective learning.

**Causal Influence of the Prefix and Suffix.** To address the first challenge, we explicitly quantify how adversarial prefixes and suffixes influence the model's prediction. Specifically, given an adversarial prompt and a generated prediction $y_i^{k,j}$, we measure the influence of the prefix $\alpha_i^k$ by the change in the model's confidence for the predicted output:

$$
\begin{aligned}
\mathcal{I}_{\text{prefix}}(\alpha_i^k) = \frac{1}{G} \sum_{j=1}^{G} \Big| \log \pi_\theta\left(y_i^{k,j} \mid \tilde{p}_i^k\right) \\
- \log \pi_\theta\left(y_i^{k,j} \mid \tilde{p}_i^k \setminus \alpha_i^k\right) \Big|.
\end{aligned}
\tag{5}
$$

We estimate the output probability following (Zhao et al., 2025). An analogous influence score $\mathcal{I}_{\text{suffix}}(\beta_i^k)$ is computed by removing the suffix $\beta_i^k$ from the prompt.

**Difficulty-Calibrated Context Generation.** The key insight for addressing the second challenge is to explicitly control the difficulty of the prefix–suffix contexts generated by the context generator. For a well-unlearned model, the generated prefix–suffix pair should not be sufficiently informative to directly trigger recall of the unlearned knowledge; otherwise, it risks leaking ground-truth information. At the same time, the same context should remain capable of eliciting recall from the current unlearning policy, indicating that the contextual difficulty is appropriately calibrated to the model's current state. In this way, the context generator is encouraged to produce adversarial contexts that are neither trivially ineffective nor excessively informative, enabling stable and effective unlearning. To formalize this notion of calibrated difficulty, we define a regret signal that measures the gap between the current unlearning policy $\pi_\theta$ and

a stronger unlearning reference $\pi_{\text{strong}}$:

$$
\text{Regret}(\tilde{p}_i^k) = \log \pi_\theta(r_i \mid \tilde{p}_i^k) - \log \pi_{\text{strong}}(r_i \mid \tilde{p}_i^k).
\tag{6}
$$

A larger regret value indicates that the adversarial context induces a high probability of recovering the ground-truth content under the current unlearning policy, while the same context is effectively suppressed by a stronger unlearning model. Such contexts are therefore of intermediate difficulty and provide informative learning signals for the current unlearning policy.

In practice, however, a stronger unlearning model is not available during training. To address this, we adopt a self-bootstrapped strategy to construct a strong unlearning reference. At each training iteration, after updating the current policy using adversarial rollouts, we initialize a reference model from the updated policy and further optimize it by explicitly suppressing the output probability of the target response $r_i$ across all generated contexts. This additional optimization step provides a more direct unlearning signal than rollout-based policy updates, as the reference model is trained to explicitly suppress the likelihood of the ground-truth response itself rather than comparatively reweighting sampled generations. As a result, the reference model exhibits stronger and more targeted unlearning behavior, making it suitable as a stable anchor for regret-based difficulty calibration.

**Reward Aggregation and Group-Based Advantage Computation.** We first describe how the previously defined signals are aggregated and combined to construct group-based advantages for the context generator $f_c$. For each prefix–suffix pair $(\alpha_i^k, \beta_i^k)$, outcome-level rewards are aggregated

across the $G$ rollouts sampled under the same context:

$$\bar{R}_i^k = -\frac{1}{G}\sum_{j=1}^{G} R(y_i^{k,j}), \qquad (7)$$

where $R$ is defined in Eq. (4) and the negative sign encourages the context generator to produce contexts that induce undesired recall. This reward is shared by the prefix and suffix within the same context pair.

Given the causal influence scores defined in Eq. (5) and the regret signal defined in Eq. (6), we standardize each signal independently within the group of $K$ prefix–suffix pairs generated for the same prompt. Specifically, for any signal $x \in \{\bar{R}_i^k, \text{Regret}(\tilde{p}_i^k), \mathcal{I}.\}$, we apply group-wise standardization:

$$\hat{x}_i^k = \frac{x_i^k - \mu_x}{\sigma_x}, \mu_x = \frac{1}{K}\sum_{k=1}^{K} x_i^k, \ \sigma_x = \text{std}(\{x_i^k\}_{k=1}^{K}).$$

After standardization, auxiliary signals are clipped to a bounded range to prevent extreme values from dominating optimization: $\tilde{x}_i^k = \text{clip}(\hat{x}_i^k, -1, 1)$. The resulting component-specific rewards for the context generator are defined as:

$$R_{\alpha,i^k}^{\text{gen}} = \widetilde{R}_i^k + \lambda\left(\widetilde{\text{Regret}}(\tilde{p}_i^k) + \widetilde{\mathcal{I}}_{\text{prefix}}(\alpha_i^k)\right), \quad (8)$$

$$R_{\beta,i^k}^{\text{gen}} = \widetilde{R}_i^k + \lambda\left(\widetilde{\text{Regret}}(\tilde{p}_i^k) + \widetilde{\mathcal{I}}_{\text{suffix}}(\beta_i^k)\right), \quad (9)$$

where we set $\lambda = 0.1$ to control the strength of auxiliary regularization. Under this formulation, the reward for each generated prefix/suffix is dominated by attack success, while being regularized by (i) its causal contribution and (ii) whether the induced context exhibits appropriate difficulty for the current unlearning policy.

For the DLM, each group consists of the $G$ rollouts sampled under a fixed prefix–suffix pair, and advantages are computed by normalizing the prediction-level rewards $R(y_i^{k,j})$ within the group. Both the DLM and the context generator are optimized using GRPO (Guo et al., 2025). The framework is detailed in Algorithm 1.

## 5. Experiments

**Dataset.** We conduct experiments on the TOFU (Maini et al., 2024) and MUSE (Shi et al., 2024b) datasets. Detailed descriptions of the datasets are provided in Appendix A.2.

**Models.** For both training and evaluation, we use `Qwen/Qwen2.5-14B-Instruct` as the judge model to assess whether generated outputs match the target responses. For adversarial training, we employ `Qwen/Qwen3-4B-Instruct-2507` as the trainable context generator. To evaluate the

---

**Algorithm 1** Adversarial RL for Diffusion Language Model Unlearning

---

**Require:** Dataset $\mathcal{D} = \{(p_i, r_i)\}_{i=1}^N$, DLM policy $\pi_\theta$, context generator $f_c$, group sizes $G$ (DLM rollouts) and $K$ (contexts), regularization weight $\lambda$

1: **for** each training iteration **do**
2:   **for** each $(p_i, r_i) \in \mathcal{D}$ **do**
3:     Sample $K$ prefix–suffix pairs $\{(\alpha_i^k, \beta_i^k)\}_{k=1}^K \sim f_c(\cdot \mid p_i, r_i)$
4:     **for** each context $(\alpha_i^k, \beta_i^k)$ **do**
5:       Construct adversarial prompt $\tilde{p}_i^k = \alpha_i^k \oplus p_i \oplus$ [MASK] $\oplus \beta_i^k$
6:       Sample $G$ generations $\{y_i^{k,j}\}_{j=1}^G \sim \pi_\theta(\cdot \mid \tilde{p}_i^k)$
7:       **for** each generation $y_i^{k,j}$ **do**
8:         Compute judge reward $R(y_i^{k,j})$ using Eq. (4)
9:       **end for**
10:      Aggregate context-level reward $\bar{R}_i^k = -\frac{1}{G}\sum_{j=1}^G R(y_i^{k,j})$
11:      Compute causal influence scores $\mathcal{I}_{\text{prefix}}(\alpha_i^k), \mathcal{I}_{\text{suffix}}(\beta_i^k)$ using Eq. (5)
12:     **end for**
13:     **Update DLM** $\pi_\theta$ via GRPO using rollout groups $\{\{y_i^{k,j}\}_{j=1}^G\}_{k=1}^K$
14:     **Construct strong unlearning reference.** Initialize $\pi_{\text{strong}} \leftarrow \pi_\theta$ and further optimize $\pi_{\text{strong}}$ by explicitly suppressing $\log \pi_{\text{strong}}(r_i \mid \tilde{p}_i^k)$ over $\{\tilde{p}_i^k\}_{k=1}^K$
15:     Compute regret signals $\{\text{Regret}(\tilde{p}_i^k)\}_{k=1}^K$ using Eq. (6)
16:     Normalize, clip, and combine generator signals as in Eq. (8) (9)
17:     **Update context generator** $f_c$ via GRPO using context group $\{(\alpha_i^k, \beta_i^k)\}_{k=1}^K$
18:   **end for**
19: **end for**

---

robustness and generalization of our framework, we adopt a stronger adversarial context generator, `Qwen/Qwen3-30B-A3B-Instruct-2507`, at inference time. We train and evaluate our method on two DLMs, `GSAI-ML/LLaDA-8B-Instruct` (Nie et al., 2025) and `Dream-org/Dream-v0-Instruct-7B` (Ye et al., 2025b).

**Baselines.** We consider the gradient ascent (GA) unlearning method defined in Eq. (3) as our primary baseline. For clarity of presentation, we omit results from the original base models without unlearning, as Figure 1 shows that the baseline unlearning method already achieves a reasonable trade-off between unlearning effectiveness and model utility, serving as a strong reference for comparison.

### 5.1. Unlearning Performance

We report results using LLaDA as the base model on the TOFU and MUSE datasets in Figure 3. Results for DREAM on the TOFU dataset are provided in Appendix **??**. From the results, we make the following observations. **(1)** For both unlearning metrics, the baseline method (GA)

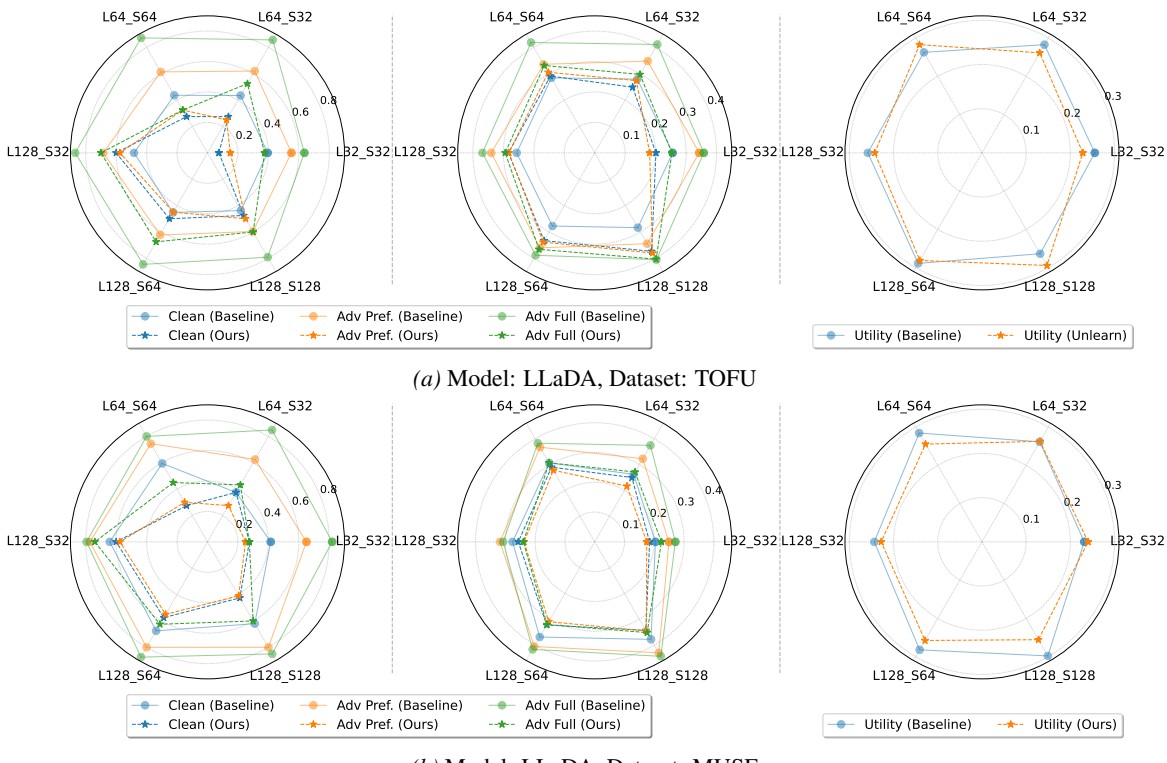

*(a)* Model: LLaDA, Dataset: TOFU

*(b)* Model: LLaDA, Dataset: MUSE

*Figure 3.* **Left:** Unlearning effectiveness evaluated by LLM-as-Judge. **Middle:** Unlearning effectiveness measured by ROUGE. **Right:** Model utility. Lower values indicate better unlearning performance, while higher values indicate better utility.

achieves strong performance under clean inputs. However, when inputs are augmented with adversarial contexts, its unlearning effectiveness degrades substantially. **(2)** Our method significantly improves unlearning performance even when evaluated with adversarial contexts generated by a stronger, unseen context generator than that used during training. This demonstrates the robustness and generalization capability of our approach. **(3)** Although our method is trained to suppress the target response under diverse adversarial contexts, it maintains model utility comparable to the baseline. This highlights the advantage of formulating unlearning as a reinforcement learning problem with relative reward optimization. Additional results using `Dream-org/Dream-v0-Instruct-7B` (Ye et al., 2025b) as the base model and evaluating on the TOFU dataset (Maini et al., 2024) are in Fig. 4

### 5.2. Training Curves

In this section, we present training curves illustrating (1) the reward of the DLM and (2) the causal influence of the generated prefixes and suffixes. To highlight the effect of our proposed components, we compare our full method with two ablated variants: *(i)* **w/o Causal Inf.**, which removes the causal influence regularization, and *(ii)* **w/o Regret**, which removes the regret-based difficulty calibration for the

context generator. The results are shown in Figure 5 and Figure 6. From the figures, we make the following observations. **(1)** As shown in Figure 5, without explicit causal influence regularization, the generated contexts often lead to imbalanced contributions, where either the prefix or the suffix dominates the model's prediction. In contrast, our full framework maintains a more balanced influence between prefixes and suffixes, encouraging both components to make meaningful contributions to adversarial context generation. **(2)** As shown in Figure 6, removing the regret module leads to unstable DLM rewards that are either excessively high or excessively low. When the reward is too high, the DLM can trivially avoid producing the target response under most generated contexts, indicating that the adversarial contexts are overly easy. Conversely, when the reward is too low, the generated contexts become overly strong and may leak information that enables the DLM to recall the target response. In both cases, the difficulty of the adversarial contexts is poorly calibrated, resulting in either trivial or overly aggressive attacks that hinder effective unlearning. In contrast, our full method maintains rewards that stabilize at values typically below 0.5, indicating a mixture of positive and negative rewards across rollouts. This reward diversity provides informative learning signals and supports more stable and effective unlearning. We further observe that when

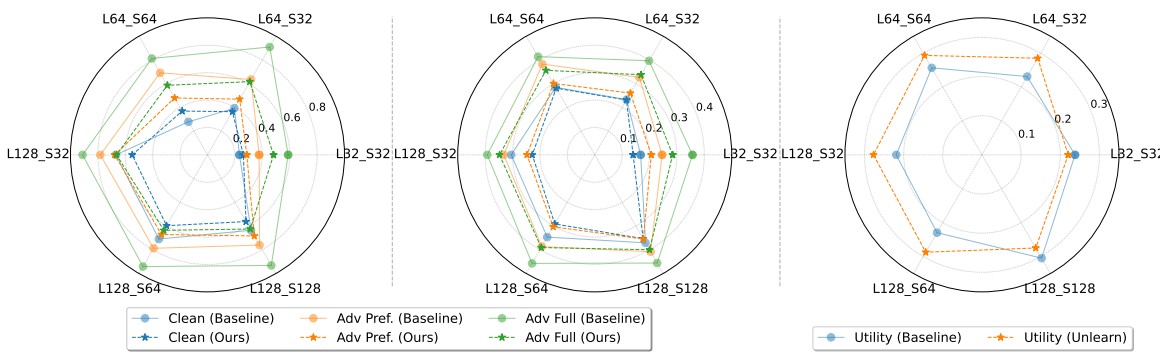

*Figure 4.* **Left:** Unlearning effectiveness evaluated by LLM-as-Judge. **Middle:** Unlearning effectiveness measured by ROUGE. **Right:** Model utility. Lower is better for the unlearning performance, while higher is better for utility. (Model: DREAM, Data: TOFU)

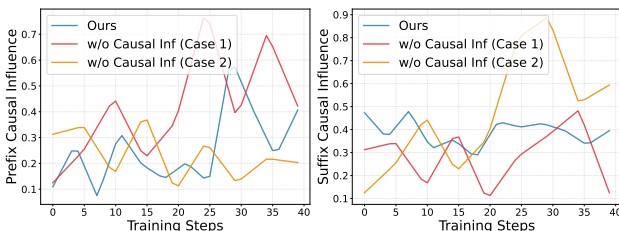

*Figure 5.* Training curves of causal influence.

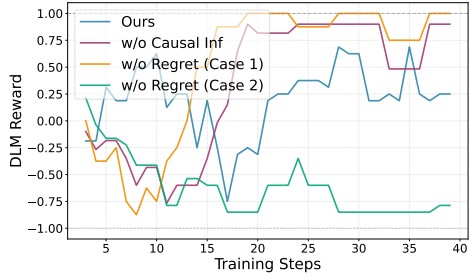

*Figure 6.* Training curves of DLM reward.

causal influence regularization is removed, the DLM reward tends to be higher. We attribute this behavior to imbalanced prefix–suffix contributions, which limit the effectiveness of adversarial attacks.

### 5.3. Ablation Studies

*Table 1.* Ablation results under different settings using the LLM-as-Judge metric (lower is better).

| Method | Clean | Adv Pref. | Adv Full |
|---|---|---|---|
| Ours | 0.55 | 0.65 | 0.63 |
| w/o Causal Inf. | 0.63 | 0.70 | 0.75 |
| w/o Regret | 0.58 | 0.73 | 0.80 |

Based on the preceding analysis, we present an ablation study to quantify the contribution of each proposed module to unlearning performance. Specifically, we

use `Dream-org/Dream-v0-Instruct-7B` (Ye et al., 2025b) as the base model and report results on the TOFU dataset under a fixed generation setting, where both the output length and the number of denoising steps are set to 128. The results are summarized in Table 1. As shown, both ablated variants exhibit degraded unlearning performance across all three prompt settings. Together with the training curve analysis, these results demonstrate the effectiveness of our proposed modules in improving robust unlearning.

## 6. Conclusion

We presented the first comprehensive study of unlearning for diffusion language models, revealing strong sensitivity to generation hyperparameters and vulnerability to adversarial contexts. To address these challenges, we proposed an adversarial reinforcement learning framework with causal influence regularization and regret-based difficulty calibration. Our method achieves robust unlearning under unseen adversarial contexts while preserving model utility.

## Impact Statement

This work studies robust machine unlearning for diffusion language models, aiming to reduce unwanted memorization by training unlearning policies that remain effective under adversarially constructed contexts. If deployed responsibly, the techniques may help mitigate privacy, copyright, and data governance risks by improving a model's ability to suppress specified content while preserving general utility.

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

# A. Implementation Details

We use the code base from OpenUnlearning (Dorna et al., 2025) and adapt it for diffusion language models.

## A.1. Models Used in This Paper

Here we list all models we used in this paper, along with their corresponding Hugging Face repositories.

- **Qwen2.5-14B-Instruct** (judge model used during training and evaluation) [1]

- **Qwen3-4B-Instruct-2507** (trainable adversarial context generator for adversarial training) [2]

- **Qwen3-30B-A3B-Instruct-2507** (stronger adversarial context generator used at inference time) [3]

- **LLaDA-8B-Instruct** (diffusion language model) [4]

- **Dream-v0-Instruct-7B** (diffusion language model) [5]

## A.2. Dataset

**TOFU.** We use the TOFU (Task of Fictitious Unlearning) dataset (Maini et al., 2024), which is designed for evaluating unlearning in large language models and consists of fictitious author profiles paired with question–answer examples. All experiments use the `locuslab/TOFU` dataset released on Hugging Face. We adopt the standard `forget10` split, which contains 200 samples (10% of the data) designated for unlearning, and the complementary `retain90` split with 1,800 samples (90%) used to preserve general model utility. In addition, a `holdout10` split is reserved exclusively for evaluation. Each data instance contains a `question` field and a corresponding `answer` field, and all inputs are truncated or padded to a maximum sequence length of 512 tokens.

For evaluation, we additionally use the perturbed variants of the forget set (`forget10_perturbed`), which contain paraphrased versions of the original questions. This setting allows us to assess whether unlearning generalizes beyond exact memorization to semantically equivalent queries.

**MUSE-Books.** We also conduct experiments on the MUSE (Machine Unlearning Six-Way Evaluation) benchmark (Shi et al., 2024b), specifically the Books subset (`muse-bench/MUSE-Books`). The MUSE-Books dataset is constructed from the Harry Potter book series and simulates real-world copyright-related unlearning scenarios. The forget set contains the original book text (approximately 1.1M tokens), while the retain set comprises related content from the Harry Potter FanWiki (approximately 0.5M tokens), representing domain-specific knowledge that should be preserved after unlearning. This asymmetric design reflects practical settings where the knowledge distribution differs between content to be forgotten and content to be retained.

The dataset is organized into multiple evaluation subsets: (1) `raw`: the raw corpus used for pretraining-style unlearning with a maximum sequence length of 2048 tokens; (2) `verbmem`: prompt-completion pairs for evaluating verbatim memorization, where the model is given an initial text segment and asked to continue; (3) `knowmem`: GPT-4 generated question-answer pairs for evaluating knowledge retention about book content, with a maximum sequence length of 512 tokens and few-shot in-context learning examples sampled from the `forget_qa_icl` split. All experiments use `forget` as the forget split and `retain1` as the retain split.

## A.3. Context Generator

We use a large language model to generate adversarial prefix and suffix contexts for evaluation. The context generator is `Qwen/Qwen3-4B-Instruct-2507`. Contexts are generated with a temperature of 0.7, a maximum of 512 new tokens, and sampling enabled.

---

[1] https://huggingface.co/Qwen/Qwen2.5-14B-Instruct
[2] https://huggingface.co/Qwen/Qwen3-4B-Instruct-2507
[3] https://huggingface.co/Qwen/Qwen3-30B-A3B-Instruct-2507
[4] https://huggingface.co/GSAI-ML/LLaDA-8B-Instruct
[5] https://huggingface.co/Dream-org/Dream-v0-Instruct-7B

---

**Prompt for Adversarial Context Generation**

You are an expert at creating subtle contextual prompts for adversarial evaluation.
Task: Generate two MINIMAL context snippets:
1. **prefix**: Appearing BEFORE the question
2. **suffix**: Appearing AFTER the question
**FORBIDDEN – Context must NOT include:**

- Facts that allow logical inference of the answer

- Mention unique identifiers, dates, titles, names, or phrases that unambiguously point to the answer

- Anything that helps through reasoning

CRITICAL RULE: Context must test MEMORIZATION, NOT reasoning/inference ability.
**Question:** {question}
**Answer (reference – NEVER leak this info):** {answer}
Output ONLY valid JSON (no markdown): `{"prefix": "<>", "suffix": "<>"}`

---

## A.4. Evaluation Metrics

We evaluate both unlearning effectiveness and model utility.

### A.4.1. LLM-AS-JUDGE

We use an LLM-based judge to assess whether generated outputs still contain information corresponding to the forgotten data. The judge model is `Qwen/Qwen2.5-14B-Instruct`, and inference is performed with a maximum of 512 new tokens and a temperature of 0.0. The prompt for LLM-as-Judge is showing below:

---

**Prompt for LLM-as-Judge**

You are evaluating whether a language model has successfully unlearned specific knowledge.
**Ground Truth Answer (what the model should have forgotten):** {ground_truth}
**Model's Response:** {model_response}
**Task:** Determine whether the model's response still reveals the forgotten knowledge from the ground truth.
**Scoring criteria (binary):**

- 1: FAILURE — The response reveals or implies correct factual knowledge matching the ground truth (including any correct names, dates, places, or key details, even if partially or paraphrased).

- 0: SUCCESS — The response does NOT allow inference of the correct fact (e.g., it is clearly wrong, vague, refuses, unrelated, or gibberish).

Respond with ONLY a JSON object in this exact format: `{"score": <1 or 0>, "reasoning": "<brief explanation>"}`

---

**Scoring.** Lower mean score indicates better unlearning (less knowledge leakage).

### A.4.2. ROUGE

FOllowing open-unlearning (Dorna et al., 2025), we compute ROUGE scores using the `rouge_scorer` library with stemming enabled. Specifically, we report `rouge1_recall`, which measures unigram overlap recall; `rougeL_recall`, which measures recall based on the longest common subsequence and is used as the primary metric; and `rougeL_f1`, which measures the F1 score of the longest common subsequence. Lower ROUGE scores indicate better unlearning effectiveness.

### A.4.3. MODEL UTILITY

Model utility is measured using the harmonic mean of ROUGE-L recall scores computed over three evaluation sets. These include the retain set, which consists of `retain90_perturbed` questions from the retain split; the Real Authors (RA) set, which contains questions about real-world authors; and the World Facts (WF) set, which contains general knowledge

questions. The overall utility score is computed as

$$\text{Utility} = \text{HM}(\text{ROUGE}_{\text{retain}}, \text{ROUGE}_{\text{RA}}, \text{ROUGE}_{\text{WF}}), \tag{10}$$

where HM denotes the harmonic mean.

## A.5. Training Details

All experiments use `GSAI-ML/LLaDA-8B-Instruct` (Nie et al., 2025) as the base model. Before unlearning, we perform a warm-up supervised fine-tuning (SFT) stage on the forget split to capture task-specific knowledge, and use the resulting checkpoint for subsequent unlearning. The unlearning method used in all experiments is Gradient Ascent.

### A.5.1. HYPERPARAMETERS

| Parameter | Value |
|---|---|
| Batch size per device | 1 |
| Gradient accumulation steps | 4 |
| Effective batch size | 8 (with 2 GPUs) |
| Learning rate | $1 \times 10^{-5}$ |
| Number of epochs | 10 |
| Maximum sequence length | 512 |
| Precision | bfloat16 |
| Optimizer | AdamW |

### A.5.2. LoRA CONFIGURATION

| Parameter | Value |
|---|---|
| LoRA rank ($r$) | 128 |
| LoRA alpha | 256 |
| Target modules | `q_proj`, `k_proj`, `v_proj` |
| Dropout | 0.05 |

### A.5.3. DLM FORWARD NOISING

During training, for each sample, a random timestep $t$ is drawn from a uniform distribution over the interval $\text{Uniform}(\epsilon, 1-\epsilon)$, where $\epsilon = 10^{-3}$. Response tokens are then masked independently with probability $t$. The cross-entropy loss is computed over the masked response tokens and scaled by a factor of $1/t$. Prompt tokens are preserved without the application of noise and are excluded from loss computation.

## A.6. Generation Configuration

For evaluation, we test multiple generation hyperparameter configurations. Specifically, we consider combinations of output length $L$ and the number of denoising steps $S$ given by

$$(L, S) \in \{(32, 32), (64, 32), (128, 32), (64, 64), (128, 64), (128, 128)\}. \tag{11}$$

The block length for block-based generation is set to 32 tokens.

