# OpenReview forum: "Adversarial Reinforcement Learning for Robust Diffusion Large Language Model Unlearning"
_ICML.cc/2026/Conference — ICML 2026 regular_

### Official Review · Reviewer_hjXv · 2026-02-21

**Soundness:** 3
**Presentation:** 2
**Significance:** 4
**Originality:** 4
**Overall Recommendation:** 5
**Confidence:** 5

**Summary:**

This paper explores machine unlearning in Diffusion Language Models (DLMs), highlighting their vulnerability to adversarial prefix/suffix contexts. To address this, the authors propose an adversarial RL framework where a context generator crafts attacks to elicit forgotten info, while the DLM is trained to suppress it. The training is stabilized using causal influence regularization and regret-based difficulty calibration, leading to improved unlearning robustness without sacrificing general utility.

**Compliance With Llm Reviewing Policy:**

Affirmed.

**Final Justification:**

My concerns have been addressed, so I’m raising my score to 5.

**Key Questions For Authors:**

See above.

**Limitations:**

1. The adversarial training pipeline using GRPO with a context generator inherently increases the computational cost. A brief analysis of this overhead is needed.
2. The method is currently optimized against prefix/suffix attacks. The authors should explicitly acknowledge the limitation that the unlearned model might still be vulnerable to other forms of discrete prompt attacks (e.g., interleaved mask-text infilling).

**Strengths And Weaknesses:**

**Strength**
1. The paper first present the unlearning in discrete diffuison, which is a really important issue.
2. This paper suggests distinguished promblem in discrete diffusion model other than autoregressive models (e.g., block size, generation length), and also suggests reasonable method as well. Definitely a solid work.

**Weakness**
1. Suffix attacks may not be sufficient. Why not use interleaved filling instead? You may refer to DIJA [1] for reference.
2. Even if there are no established methods for dLLMs, the baseline might be too simple. Can you add comparisons to methods like NPO [2] (if possible)? Also, there are some methods working for safety alignment specialized for dLLMs, such as A2D [3], that could be considered for unlearning.
3. Your method may introduce significant computational overhead. Could you provide an analysis of this?

I am willing to raise my score if my concerns are adequately resolved.

[1] The devil behind the mask: An emergent safety vulnerability of diffusion llms. ICLR (2026).\
[2] Negative preference optimization: From catastrophic collapse to effective unlearning. COLM (2024).\
[3] A2d: Any-order, any-step safety alignment for diffusion language models. ICLR (2026).

---

> ### Author Rebuttal · Authors · 2026-03-31
>
> We sincerely thank the reviewer for the support on our paper.
>
> ----
>
> ## W1 & Limitation 2: Suffix attacks may not be sufficient. Why not use interleaved filling instead?
> We agree that interleaved filling is an important and stronger DLM-specific attack family. However, in our setting, the forgotten targets are typically short QA-style answers, which makes it difficult to construct meaningful interleaved mask-text attacks without directly revealing substantial parts of the target answer itself. Our goal is to test whether forgotten knowledge can still be recovered from informative surrounding context, rather than from partial answer disclosure. For this reason, we focus on adversarial prefix/suffix contexts, which provide a cleaner and more controllable evaluation setting for the short-answer QA-style unlearning setup considered in this paper.
>
> To further address the reviewer’s concern, we additionally follow DIJA[1] to test whether our framework transfers to a stronger jailbreak setting. Specifically, we first fine-tune the victim DLM with our method, where we adopt interleaved context generation, and then attack the fine-tuned model using DIJA. We report results on HarmBench and JailbreakBench:
>
>
> | Benchmark | Metric | DIJA | Ours + attacked by DIJA |
> |-|-|-:|-:|
> | HarmBench | ASR-e | 60.0 | 33.6 |
> | JailbreakBench | ASR-e | 81.0 | 42.0 |
>
> These results show that even under the stronger DIJA jailbreak setting, our method still substantially reduces ASR-e.
>
> We sincerely thank the reviewer for pointing this out, and we will include this discussion in the revised version.
>
> ---------
>
> ## W2 & W3 & Limitation1: more baselines & discussion on computation overhead
> We thank the reviewer for highlighting both the need for stronger baselines and the importance of discussing the computational overhead of our framework.
>
> To address both concerns jointly, we compare against additional baselines under varying rollout budgets. The main extra cost of our method comes from the enlarged **DLM-side rollout budget** induced by adversarial context expansion, while auxiliary modules contribute a smaller fraction of the overhead. Concretely, for each prompt, we generate `K` adversarial contexts and perform `G` rollouts per context. We compare different methods under the same `(K, G)` setting, so they operate under comparable rollout budgets.
>
> ### Additional Baselines
>
> -   **NPO**: We replace its AR preference objective with the corresponding DLM objective.
>
> -   **A2D**: We adopt A2D[1] by reversing its gradient direction for the unlearning objective.
>
> -   **Quark**: We adapt Quark[2] as a reward-conditioned DLM baseline using the same judge and KL anchoring.
>
>
> Under this compute-matched DLM setting, we report the results with setting (L=64, S=32):
>
>
> | Method |    K=1, G=4 |    K=2, G=4 |    K=3, G=4 |    K=4, G=4 |    K=4, G=8 |
> | - | -: | -: | -: | -: | -: |
> | GA     | 0.80 / 0.29 | 0.77 / 0.24 | 0.90 / 0.11 | 0.91 / 0.06 | 0.94 / 0.04 |
> | NPO    | 0.78 / 0.27 | 0.75 / 0.19 | 0.75 / 0.20 | 0.80 / 0.16 | 0.83 / 0.13 |
> | A2D    | 0.76 / 0.28 | 0.71 / 0.21 | 0.74 / 0.16 | 0.80 / 0.11 | 0.84 / 0.08 |
> | Quark  | 0.78 / 0.29 | 0.73 / 0.27 | 0.70 / 0.25 | 0.69 / 0.22 | 0.68 / 0.19 |
> | Ours   | 0.72 / 0.28 | 0.67 / 0.27 | 0.60 / 0.27 | 0.54 / 0.26 | 0.52 / 0.25 |
>
>
>
> Each entry is reported as Unlearn (Adv-Full) / Utility. **lower unlearning score is better**, while **higher utility is better**.
>
> GA benefits slightly from small increases in adversarial budget with fixed contexts; however, as `K` grows, context–model mismatch becomes more severe, leading to degeneration, sharp utility drop, and eventually worse Adv-Full performance. NPO is more stable than GA, but still relies on fixed contexts. A2D differs from NPO mainly in that it acts more directly at the step level during DLM generation. This can improve suppression at small `K`, but also makes it more prone to utility degradation. Quark is an RL-style baseline here. KL anchoring makes it more stable than GA/NPO, but without an online adversary or explicit difficulty calibration, its gains saturate earlier than ours.
>
> In contrast, our method benefits most consistently from larger adversarial budgets.
>
> ----
>
> [1] The Devil behind the mask: An emergent safety vulnerability of Diffusion LLMs. ICLR 26

---

> > ### Author Rebuttal · Reviewer_hjXv · 2026-04-01
> >
> > I thank the authors for the detailed rebuttal. The clarification on prefix/suffix attacks and the additional DIJA evaluation are convincing. I also appreciate the inclusion of A2D [1] and Quark [2] as adapted baselines. Given that this is the first unlearning work for DLMs, comparing against existing alignment approaches specialized for DLMs is a valuable and important contribution. I encourage the authors to include these results in the main paper. I raise my score to Accept.
> >
> > [1] Any-order, any-step safety alignment for diffusion language models.\
> > [2] Quark: Controllable text generation with reinforced unlearning.

---

> > > ### Author Response · Authors · 2026-04-01
> > >
> > > Dear Reviewer hjXv:
> > >
> > > We sincerely thank the reviewer for the thoughtful follow-up and for acknowledging our rebuttal. We are deeply encouraged by your positive assessment and truly appreciate your constructive feedback. We will incorporate these clarifications and additional results in the revised version.
> > >
> > > Sincerely,
> > >
> > > The Authors

---

### Official Review · Reviewer_T28m · 2026-03-08

**Soundness:** 3
**Presentation:** 3
**Significance:** 4
**Originality:** 4
**Overall Recommendation:** 5
**Confidence:** 3

**Summary:**

This paper addresses the issue of unlearning for diffusion language models (DLMs). It presents two primary contributions. First, it analyzes a method for unlearning and demonstrates its sensitivity to the hyperparameters of length and number of denoising steps. Second, it introduces an adversarial reinforcement learning method for DLM unlearning, describing and then addressing the key challenges of credit assignment and reward collapse that arise from a simpler RL formulation.

**Compliance With Llm Reviewing Policy:**

Affirmed.

**Final Justification:**

I am satisfied with the responses to my questions and believe this paper should be accepted.

**Key Questions For Authors:**

Why was the Qwen family of models chosen? It's a reasonable choice, but would benefit from explicit justification. It would be interesting to see how LLM-as-a-Judge-based metrics vary across model families and with the most recent LMs (assuming token cost is not prohibitive).

Did you experiment with any other approaches to difficulty calibration? Why was the presented approach selected?

**Limitations:**

Yes

**Strengths And Weaknesses:**

**Strengths**
The submission's methods are technically reasonable. The experiments use a single model family, Qwen; however, given its prevalence for a broad range of research applications, this seems sufficient to support their findings. The inclusion of ablations for the two RL methods substantially strengthens the justification for their use.

The submission is clearly written. It makes effective use of figures. Reproducing these results would require the details provided in the lengthy appendix, which is reasonable given the number of components (LLM-as-a-judge, LoRA decomposition, etc.).

The problem addressed by this paper, assessing and improving methods for unlearning in DLMs, is important to these models' safe use.

The reinforcement learning method proposed makes principled use of several RL insights: reward based on group advantage, use of LLM-as-a-judge for assessment, and adversarial task design. Although none of these components is itself novel, their combination for this particular application is original and well-designed.

**Weaknesses**

Revisions to these items would strengthen the paper, but are not strictly necessary for its acceptance.

The paper provides limited justification for the two RL adaptations proposed, i.e., measurement of causal influence of prefix and suffix to enable finer-grained rewards and difficulty calibration of prefix-suffix-contexts to control difficulty level and avoid reward collapse. In particular, no work is cited to support the latter's formulation. Given the novelty of the work, this is not unreasonable; however, this section would then benefit from additional empirics. As reward collapse is such a significant and widespread challenge in RL, such discussion and results would benefit a broad audience. The key detail here needing justification is the use of self-bootstrapping to create the strong unlearning reference. Although explicitly suppressing the target's output probability across all contexts is certainly reasonable, the paper would be strengthened by an empirical comparison (even on a very small scale) of this model's unlearning to the policy being trained. I note that the ablation study certainly supports the efficacy of this method.

A single unlearning method is presented in 3.1, which is used for the results demonstrating its hyperparameter sensitivity. There is a lack of literature on unlearning for DLMs; however, there is a substantial body of research on unlearning for text-to-image DMs. This literature may provide additional support for some of the design decisions made in the paper.

The paper would also benefit by slightly more detailed discussion of the radar plots, which are helpful and thorough but somewhat difficult to interpret without fairly extensive reading.

---

> ### Author Rebuttal · Authors · 2026-03-31
>
> We sincerely thank the reviewer for the support on our paper.
>
> ----
>
> ## (1) Justification for the two proposed RL-specific designs
> >### causal influence regularization
>
> Our motivation is the credit-assignment issue known in multi-agent RL: with only a shared reward, one component may dominate while the other becomes uninformative. In our setting, the generator produces both prefix and suffix, but without component-specific credit assignment, the reward only reflects whether the overall attack succeeds. This can cause the generator to over-rely on one side and weaken joint context generation. Our ablation already supports this point: without causal influence regularization, prefix/suffix contributions become less balanced and robustness degrades.
>
>
> >### difficulty calibration / self-bootstrapped strong reference
>
> The motivation is similar to PAIRED: useful adversarial contexts should be of intermediate difficulty, not arbitrarily hard. Directly training an additional strong policy is impractical for large DLMs, so we instead use a self-bootstrapped strong reference initialized from the current policy and then further optimized by directly suppressing the target response probability across generated contexts. This gives a stronger and more explicit unlearning signal than the preference-style update used for the main policy, while remaining computationally practical. Thus, the reference is “strong” because it receives a stronger target-suppression objective.
>
> We also empirically verify this design by comparing the target-probability ratio between the current policy and the strong reference during training:
>
>
> | Step                                  |     5 |    10 |    15 |    20 |    25 |    30 |    35 |    40 |
> | ------------------------------------- | ----: | ----: | ----: | ----: | ----: | ----: | ----: | ----: |
> | Policy / Strong Ref target-prob ratio | 1.22x | 1.51x | 2.34x | 1.62x | 2.29x | 1.81x | 1.56x | 1.35x |
>
> The ratio stays above 1 throughout, showing that the strong reference consistently assigns lower probability to the forgotten target and therefore provides a meaningful anchor for difficulty calibration.
>
> -------
>
> ## (2) Related work on text-to-image diffusion unlearning
>
> We agree this literature is relevant and will expand the discussion. However, many text-to-image diffusion unlearning methods do not transfer naturally to DLM text generation, because they rely on image-specific intervention mechanisms. For example, ESD aligns conditional outputs toward unconditional outputs to erase a concept, which is natural in text-to-image diffusion but does not have a clear counterpart in QA-style DLM generation: removing the prompt does not produce a well-defined “unconditional answer” target. More broadly, these methods are usually designed for concept/style suppression under standard prompting, rather than robust forgetting under adversarial contextual recovery. For empirical comparison, we test NPO and re-run the preliminary experiments.
>
> | Method |     L32/S32 |     L64/S32 |    L128/S32 |     L64/S64 |    L128/S64 |   L128/S128 |
> | ------ | ----------: | ----------: | ----------: | ----------: | ----------: | ----------: |
> | NPO    | 0.76 / 0.26 | 0.80 / 0.26 | 0.87 / 0.23 | 0.78 / 0.28 | 0.84 / 0.25 | 0.86 / 0.24 |
>
> We observe that NPO also remains vulnerable to adversarial context even after unlearning.
>
> ----------
>
> ## (3) Radar plots
> In the revised version, we will add a clearer explanation of how to read these plots.
>
> -----------
>
> ## (4) Why Qwen?
> We chose the Qwen family mainly for practical reasons: it provides strong, stable, open-weight models at multiple scales, making it a reproducible choice for both the generator and judge. Due to budget constraints, we are currently unable to extensively test frontier closed-source models. That said, the judge in our setting performs a relatively simple task—determining whether the forgotten target content is still recovered—rather than requiring highly open-ended reasoning. We therefore believe the current judge setup is a reasonable and practical choice for this study.
>
> To test judge sensitivity, we additionally evaluated **GPT-OSS-20B** as the judge.
>
> | Method                   | L32/S32 | L64/S32 | L128/S32 | L64/S64 | L128/S64 | L128/S128 |
> | ------------------------ | ------: | ------: | -------: | ------: | -------: | --------: |
> | GA  |    0.68 |    0.75 |     0.82 |    0.72 |     0.79 |      0.80 |
> | Ours |    0.28 |    0.37 |     0.48 |    0.33 |     0.44 |      0.46 |
>
> Although the absolute score scale shifts, the overall conclusion remains unchanged: our method still consistently outperforms GA.
>
> ---------
>
> ## (5) alternative difficulty calibration method
> We thank the reviewer for this important question. As discussed, our current design is a deliberate and practical choice. Due to rebuttal-time constraints, we did not systematically explore additional calibration strategies, but we agree this is an interesting direction for future work.

---

> > ### Author Rebuttal · Reviewer_T28m · 2026-04-01
> >
> > Thank you to the authors for their detailed responses! I would like to see the judge sensitivity experiments in the final paper. I maintain my belief that this work addresses an important problem with a sound method and should be accepted.

---

> > > ### Author Response · Authors · 2026-04-01
> > >
> > > Dear Reviewer T28m,
> > >
> > > We sincerely thank you for your thoughtful feedback and positive assessment. We are deeply encouraged by your support and truly appreciate your suggestion to include the judge sensitivity experiments, which we will incorporate in the final paper. **We would greatly appreciate it if you would consider increasing your rating to support our paper.**
> > >
> > > Sincerely,
> > >
> > > The Authors

---

### Official Review · Reviewer_rDmk · 2026-03-09

**Soundness:** 3
**Presentation:** 3
**Significance:** 3
**Originality:** 2
**Overall Recommendation:** 3
**Confidence:** 4

**Summary:**

This paper studies the machine unlearning problem in diffusion language models (DLMs) and proposes a robust unlearning framework based on adversarial reinforcement learning. The authors first conduct systematic experiments and find that the unlearning effect of DLMs is highly sensitive to generation hyperparameters and that forgotten knowledge is easily recovered when informative context is included. To address this issue, the paper proposes an adversarial training framework in which a context generator attempts to induce the model to recover the target knowledge, while the DLM learns to suppress this recovery behavior.

**Compliance With Llm Reviewing Policy:**

Affirmed.

**Final Justification:**

Although the rebuttal has partially resolved my concerns, I still believe that the novelty and overall contribution of the paper are relatively limited and I maintain my score.

**Key Questions For Authors:**

1. The paper mainly compares the proposed method with the Gradient Ascent method. However, in recent years, various approaches have been proposed in the field of machine forgetting.
2. The proposed method consists of a context generator, a discriminative model, and a reinforcement learning training process, which may theoretically incur high computational overhead.
3. The context generator can access the target response information, which may lead to the generated attack context being overly idealized.
4. Add more metrics, such as attack success rate or knowledge recovery probability.

**Limitations:**

Applicability to larger-scale diffusion language models and different generation tasks?

**Strengths And Weaknesses:**

The paper focuses on the emerging problem of forgetting in diffusion language models and reveals the unique challenges of DLMs in forgetting tasks through systematic experiments. It also presents an adversarial reinforcement learning framework and its stabilization mechanism.

---

> ### Author Rebuttal · Authors · 2026-03-31
>
> ## W1&W2: more baselins & efficiency
>
> Here we address both concerns jointly, we compare against additional baselines under varying rollout budgets. The main extra cost of our method comes from the enlarged **DLM-side rollout budget** induced by adversarial context expansion, while auxiliary modules contribute a smaller fraction of the overhead. Concretely, for each prompt, we generate `K` adversarial contexts and perform `G` rollouts per context. We compare different methods under the same `(K, G)` setting, so they operate under comparable rollout budgets.
>
> ### Additional Baselines
>
> -   **NPO**: We replace its AR preference objective with the corresponding DLM objective.
>
> -   **A2D**: We adopt A2D[1] by reversing its gradient direction for the unlearning objective.
>
> -   **Quark**: We adapt Quark[2] as a reward-conditioned DLM baseline using the same judge and KL anchoring.
>
> Under this compute-matched DLM setting, we report the results with setting (L=64, S=32):
>
> | Method |    K=1, G=4 |    K=2, G=4 |    K=3, G=4 |    K=4, G=4 |    K=4, G=8 |
> | - | -: | -: | -: | -: | -: |
> | GA     | 0.80 / 0.29 | 0.77 / 0.24 | 0.90 / 0.11 | 0.91 / 0.06 | 0.94 / 0.04 |
> | NPO    | 0.78 / 0.27 | 0.75 / 0.19 | 0.75 / 0.20 | 0.80 / 0.16 | 0.83 / 0.13 |
> | A2D    | 0.76 / 0.28 | 0.71 / 0.21 | 0.74 / 0.16 | 0.80 / 0.11 | 0.84 / 0.08 |
> | Quark  | 0.78 / 0.29 | 0.73 / 0.27 | 0.70 / 0.25 | 0.69 / 0.22 | 0.68 / 0.19 |
> | Ours   | 0.72 / 0.28 | 0.67 / 0.27 | 0.60 / 0.27 | 0.54 / 0.26 | 0.52 / 0.25 |
>
> Each entry is reported as Unlearn (Adv-Full) / Utility. **lower unlearning score is better**, while **higher utility is better**.
>
> GA benefits slightly from small increases in adversarial budget with fixed contexts; however, as `K` grows, context–model mismatch becomes more severe, leading to degeneration, sharp utility drop, and eventually worse Adv-Full performance. NPO is more stable than GA, but still relies on fixed contexts. A2D differs from NPO mainly in that it acts more directly at the step level during DLM generation. This can improve suppression at small `K`, but also makes it more prone to utility degradation. Quark is an RL-style baseline here. KL anchoring makes it more stable than GA/NPO, but without an online adversary or explicit difficulty calibration, its gains saturate earlier than ours.
>
> In contrast, our method benefits most consistently from larger adversarial budgets.
>
> ---
>
> ## W3:context generator can access the target response
>
> We agree that access to the target response may make the generated contexts overly idealized. **This is exactly why we introduce regret-based difficulty calibration**: instead of rewarding arbitrarily strong attacks, we encourage the generator to produce contexts of intermediate difficulty—challenging for the current unlearning model, but not trivially answer-revealing.
>
> ---
>
> ## W4: more metrics: ASR and recovery probability
>
> (1) **Our LLM-as-Judge already acts as a judge-based ASR metric.**
>
> (2) We additionally report KRP and ASR@4 under the six generation settings. For each prompt, we sample 4 outputs and use the same LLM judge to check whether the forgotten target is recovered. KRP is the average recovery fraction over 4 samples, and ASR@4 indicates whether recovery occurs at least once. Lower is better for both.
>
> | Method |     L32/S32 |     L64/S32 |    L128/S32 |     L64/S64 |    L128/S64 |   L128/S128 |
> | - | -: | -: | -: | -: | -: | -: |
> | GA     | 0.81 / 0.93 | 0.86 / 0.94 | 0.92 / 0.97 | 0.84 / 0.93 | 0.88 / 0.96 | 0.90 / 0.97 |
> | Ours   | 0.47 / 0.65 | 0.54 / 0.62 | 0.50 / 0.60 | 0.51 / 0.68 | 0.47 / 0.56 | 0.49 / 0.59 |
>
> Each entry is reported as KRP / ASR@4. Our method consistently yields substantially lower recovery than GA.
>
> ---
>
> ## Limitation 1: larger DLM & other generation tasks
>
> >### larger DLM
> We additionally test on inclusionAI/LLaDA2.0-mini (16B). Due to compute constraints, we do not test 100B-scale models.
>
> | Method | L32/S32 | L64/S32 | L128/S32 | L64/S64 | L128/S64 | L128/S128 |
> | - | -: | -: | -: | -: | -: | -: |
> | GA     |    0.84 |    0.85 |     0.89 |    0.83 |     0.87 |      0.88 |
> | Ours   |    0.30 |    0.38 |     0.55 |    0.35 |     0.49 |      0.53 |
>
> The same trend holds: GA remains highly vulnerable, while our method remains substantially stronger.
>
> >### more tasks
>
> We also follow DIJA[3] to test transfer to a jailbreak setting by first fine-tuning the victim DLM with our method and then attacking it with DIJA. We report results on **HarmBench** and **JailbreakBench**.
>
> | Benchmark | Metric | DIJA | Ours + attacked by DIJA |
> |-|-|-:|-:|
> | Harm| ASR-e | 60.0 | 33.6 |
> | Jailbreak| ASR-e | 81.0 | 42.0 |
>
> Even under the stronger DIJA jailbreak setting, our method substantially reduces ASR-e.
>
> ---
>
> [1] A2D: Any-Order, Any-Step Safety Alignment for Diffusion Language Models.ICLR 26
>
> [2] Quark: Controllable Text Generation with Reinforced [Un]learning.NeurIPS 22
>
> [3] The Devil behind the mask: An emergent safety vulnerability of Diffusion LLMs.ICLR 26

---

> > ### Author Rebuttal · Reviewer_rDmk · 2026-04-03
> >
> > Although the authors provide a comparison by adjusting rollout budgets, the authors failed to respond to complexity Weaknesses 2. Therefore, I maintain my score.

---

> > > ### Author Response · Authors · 2026-04-05
> > >
> > > Dear Reviewer rDmk,
> > >
> > > We sincerely thank you for the continued discussion.
> > >
> > > To clarify, the main additional cost of our framework compared with simpler baselines comes from the enlarged rollout budget induced by adversarial context expansion. For this reason, in our rebuttal we reported compute-matched comparisons, showing that **under the same rollout budget our method consistently achieves the strongest robustness-performance tradeoff among the compared baselines**.
> > >
> > > Regarding the **context generator**, as discussed in the paper, forgotten knowledge in DLMs can often be recovered through informative prefix-suffix contexts, and in practice there can be many such effective context pairs. To make the unlearned model robust, it is therefore important to expose it to diverse attacking contexts rather than relying on a fixed small set. **Whether these contexts are generated offline in advance or online during training, one must still pay the cost of producing diverse attack contexts**;
> > >
> > > As for the **discriminative model**, we use an LLM-as-judge because metrics such as ROUGE are often insufficient to determine whether the forgotten target has been semantically recovered. The judge does not constitute a training-time optimization burden. More broadly, **In the era of LLMs, LLM-as-judge is already a widely adopted evaluation tool in research [1]**.
> > >
> > > Finally, regarding **reinforcement learning**, we note that competing unlearning baselines, whether based on reinforcement learning or supervised fine-tuning, also optimize and update the model parameters. In this sense, **optimization itself is not an additional source of complexity unique to our method**.
> > >
> > > Overall, we believe the complexity-performance tradeoff of our framework is justified. More importantly, as one of the first works to systematically study unlearning in diffusion language models and to propose a robust unlearning framework tailored to this setting, we hope the reviewer will also take the significance of this contribution into consideration.
> > >
> > > We appreciate your feedback and will make these clarifications more explicit in the revised version.
> > >
> > > Sincerely,
> > >
> > > The Authors
> > >
> > > -----
> > >
> > > [1] A Survey on LLM-as-a-Judge. arXiv:2411.15594

---

### Official Review · Reviewer_tPYZ · 2026-03-13

**Soundness:** 3
**Presentation:** 3
**Significance:** 3
**Originality:** 2
**Overall Recommendation:** 4
**Confidence:** 3

**Summary:**

This paper presents a study on machine unlearning specifically tailored for diffusion language models. The authors demonstrate that dLLMs are highly sensitive to generation hyperparameters and vulnerable to adversarial contexts (such as prefix and suffix injections) that can elicit supposedly unlearned knowledge. To mitigate this, the paper proposes an adversarial reinforcement learning algorithm.

**Compliance With Llm Reviewing Policy:**

Affirmed.

**Final Justification:**

My concerns have been addressed, so I’m raising my score to 4.

**Key Questions For Authors:**

See Weaknesses.

**Limitations:**

No.

**Strengths And Weaknesses:**

**Strengths:**
1. This is one of the first comprehensive studies to explore the mechanics and vulnerabilities of machine unlearning specifically within diffusion language models.
2. The introduction of causal influence regularization and regret-based difficulty calibration effectively addresses the instability issues.

**Weaknesses:**
1. The observation that diffusion models are susceptible to suffix or interleaved context manipulation is not entirely new. This specific vulnerability has already been highlighted in recent safety literature, such as Wen et al., "The Devil behind the mask: An emergent safety vulnerability of Diffusion LLMs".
2. Framing unlearning as a reinforcement learning or preference optimization problem is already well-established in the autoregressive LLM literature. Methods such as Quark (Lu et al., 2022), DeMem (Kassem et al., 2023), DPO-based unlearning (Rafailov et al., 2023), and NPO (Zhang et al., 2024) have extensively explored this paradigm.
3. The proposed framework is only compared against a basic Gradient Ascent baseline. To prove the superiority of the method, it is crucial to compare it against other state-of-the-art RL-based or preference-based unlearning methods adapted for DLMs.
4. The adversarial RL framework requires a separate context generator (e.g., a 4B parameter model) and multiple rollouts per training step. This introduces significant computational and memory overhead compared to standard unlearning methods. The paper lacks a quantitative discussion on the training efficiency and cost trade-offs.

---

> ### Author Rebuttal · Authors · 2026-03-31
>
> We sincerely thank the reviewer for the constructive and helpful feedback.
>
> ----
>
> ## W1:The observation is not entirely new.
>
> We thank the reviewer for this helpful observation. We do cite [1] in the introduction, and we apologize if the distinction was not sufficiently clear. Our contribution differs in two key respects:
>
> (1) **Fixed-model vulnerability vs. post-fine-tuning robustness**.
>
> Wen et al. study a fixed DLM and show that it is vulnerable to interleaved mask-text jailbreak prompts. In contrast, we ask whether a DLM that has already been fine-tuned for unlearning or trained with adversarial contexts remains vulnerable。
>
> (2) **Identifying the vulnerability vs. training against it**.
>
> Wen et al. reveal the existence of this vulnerability, but do not study whether a fine-tuned DLM can be made robust to such attacks, nor how to train a DLM to robustly forget undesired information under adversarial contexts.
>
> -----
>
> ## W2: RL for unlearning is well-established
>
> We agree that framing unlearning as reinforcement learning or preference optimization is not new in the autoregressive LLM literature. **However, this is not the main novelty of our paper.**
>
> Instead, our contribution is to study robust unlearning in DLMs under adversarial contextual attacks. We show that combining unlearning with adversarial context generation introduces new challenges, including prefix/suffix credit assignment ambiguity and difficulty mismatch between the adversary and the unlearning model. To address these issues, we propose a dedicated framework with causal influence regularization and regret-based difficulty calibration.
>
> ----
>
> ## W3 & W4: comparison to more baselines & discussion on the efficiency and cost trade-offs
>
> We thank the reviewer for highlighting both the need for stronger baselines and the importance of discussing the computational overhead of our framework.
>
> To address both concerns jointly, we compare against additional baselines under varying rollout budgets. The main extra cost of our method comes from the enlarged **DLM-side rollout budget** induced by adversarial context expansion, while auxiliary modules contribute a smaller fraction of the overhead. Concretely, for each prompt, we generate `K` adversarial contexts and perform `G` rollouts per context. We compare different methods under the same `(K, G)` setting, so they operate under comparable rollout budgets.
>
> ### **Additional Baselines**
>
> -   **NPO**: We replace its AR preference objective with the corresponding DLM objective.
>
> -   **A2D**: We adopt A2D[2] by reversing its gradient direction for the unlearning objective.
>
> -   **Quark**: We adapt Quark[3] as a reward-conditioned DLM baseline using the same judge and KL anchoring.
>
>
> Under this compute-matched DLM setting, we report the results with setting (L=64, S=32):
>
>
> | Method |    K=1, G=4 |    K=2, G=4 |    K=3, G=4 |    K=4, G=4 |    K=4, G=8 |
> | ------ | ----------: | ----------: | ----------: | ----------: | ----------: |
> | GA     | 0.80 / 0.29 | 0.77 / 0.24 | 0.90 / 0.11 | 0.91 / 0.06 | 0.94 / 0.04 |
> | NPO    | 0.78 / 0.27 | 0.75 / 0.19 | 0.75 / 0.20 | 0.80 / 0.16 | 0.83 / 0.13 |
> | A2D    | 0.76 / 0.28 | 0.71 / 0.21 | 0.74 / 0.16 | 0.80 / 0.11 | 0.84 / 0.08 |
> | Quark  | 0.78 / 0.29 | 0.73 / 0.27 | 0.70 / 0.25 | 0.69 / 0.22 | 0.68 / 0.19 |
> | Ours   | 0.72 / 0.28 | 0.67 / 0.27 | 0.60 / 0.27 | 0.54 / 0.26 | 0.52 / 0.25 |
>
>
>
> Each entry is reported as Unlearning (Adv-Full) / Utility. **lower unlearning score is better**, while **higher utility is better**.
>
> GA benefits slightly from small increases in adversarial budget with fixed contexts; however, as `K` grows, context–model mismatch becomes more severe, leading to degeneration, sharp utility drop, and eventually worse Adv-Full performance. NPO is more stable than GA, but still relies on fixed contexts. A2D differs from NPO mainly in that it acts more directly at the step level during DLM generation. This can improve suppression at small `K`, but also makes it more prone to utility degradation. Quark is an RL-style baseline here. KL anchoring makes it more stable than GA/NPO, but without an online adversary or explicit difficulty calibration, its gains saturate earlier than ours.
>
> In contrast, our method benefits most consistently from larger adversarial budgets because it combines online adversarial context optimization with difficulty calibration. This not only avoids the fixed-context mismatch of the other baselines, but also keeps the generated attacks at a useful difficulty level, leading to more stable and effective training.
>
> Overall, these results suggest that under a similar budget, **baselines do not obtain comparable gains, whereas our method does**.
>
> --------------
>
> [1] The Devil behind the mask: An emergent safety vulnerability of Diffusion LLMs. ICLR 26
>
> [2] A2D: Any-Order, Any-Step Safety Alignment for Diffusion Language Models. ICLR 26
>
> [3] Quark: Controllable Text Generation with Reinforced [Un]learning. NeurIPS 22.

---

> > ### Author Rebuttal · Reviewer_tPYZ · 2026-04-03
> >
> > Thank you for the detailed response. My concerns have been addressed, so I’m raising my score to 4.

---

> > > ### Author Response · Authors · 2026-04-05
> > >
> > > Dear Reviewer tPYZ,
> > >
> > > We sincerely thank you for acknowledging our rebuttal and for your support of our paper. We truly appreciate your thoughtful and constructive feedback.
> > >
> > > Sincerely,
> > > The Authors

---

### Decision · Program_Chairs · 2026-04-30

**Decision:**

Accept (regular)

**Comment:**

This paper addresses machine unlearning in diffusion language models (DLMs). The submission shows that unlearning in DLMs is sensitive to decoding configurations and can be easily undermined by contextual attacks. To improve robustness, the authors propose an adversarial reinforcement learning framework that jointly trains a context generator and the target model, with additional mechanisms to improve reward assignment and training stability.

A key strength of the paper is that it tackles a genuinely important gap in the literature by examining unlearning in DLMs rather than standard autoregressive models. The proposed training framework is technically reasonable and thoughtfully designed for this setting, and the experiments are generally well executed.

The main limitations concern novelty and efficiency. While the overall application to DLM unlearning is new, several individual ingredients of the proposed framework draw from existing reinforcement learning and adversarial training ideas, so the methodological advance is more in the adaptation and integration than in fundamentally new algorithmic concepts. In addition, the method introduces additional training overhead through adversarial context generation and expanded rollouts.

However, the rebuttal addressed most substantive concerns and materially improved the strength of this work. Most importantly, the paper makes a meaningful contribution by opening up robust unlearning for diffusion language models and providing a technically sound solution. There are still some remaining issues as reflected by the 3-scored reviewer, yet, they are not fundamental flaws. On balance, I believe the paper merits acceptance.